# Research on Preventing High-Density Materials from Settling in Liquid Resin

**DOI:** 10.3390/ma18153469

**Published:** 2025-07-24

**Authors:** Lixin Xuan, Zhiqiang Wang, Xuan Yang, Xiao Wu, Junjiao Yang, Shijun Zheng

**Affiliations:** 1AVIC Research Institute for Special Structures of Aeronautical Composite, Jinan 250023, China; 2College of Chemistry, Beijing University of Chemical Technology, Beijing 100029, China; 3College of Materials Science, Beijing University of Chemical Technology, Beijing 100029, China

**Keywords:** magnetic particles, hollow silica, surface coating

## Abstract

The applications of magnetic particles in anti-counterfeiting and anti-absorbing coatings and other functional materials are becoming increasingly widespread. However, due to their high density, the magnetic particles rapidly settle in organic resin media, significantly affecting the quality of the related products. Thereby, reducing the density of the particles is essential. To achieve this goal, high-density magnetic particles were coated onto the surface of hollow silica using anion–cation composite technology. Further, the silane coupling agent N-[3-(trimethoxysilyl)propyl]ethylenediamine was bonded to the surface of magnetic particles to form an amino-covered interfacial layer with a pH value of 9.28, while acrylic acid was polymerized and coated onto the surface of hollow silica to form a carboxyl-covered interfacial layer with a pH value of 4.65. Subsequently, the two materials were compounded to obtain a low-density composite magnetic material. The morphologies and structural compositions of the magnetic composite materials were studied by FTIR, SEM, SEM-EDS, XRD, and other methods. The packing densities of the magnetic composite materials were compared using the particle packing method, thereby solving the problem of magnetic particles settling in the resin solution.

## 1. Introduction

Magnetic materials play an important role in daily life and the defense industry. They are important components in energy devices such as motors, generators, transformers, and actuators [1]. The growth and development of magnetic materials have led to many innovative applications, including the applicability of magnetic fluids, the labeling and classification of biological species, biomedical imaging, site-specific drug delivery, and magnetic cell lysis [2]. Magnetic materials also play an important role in the absorption and attenuation of electromagnetic waves [3]. This property plays a vital role in the absorption efficiency of electromagnetic-wave-absorbing materials, which provide an effective means of solving electromagnetic radiation problems by converting that energy into other forms of energy to achieve dissipation. In particular, iron-based magnetic absorbers exhibit strong magnetic loss characteristics, such as high magnetic permeability and saturation magnetization, and are widely used in fields such as information communication, electronic equipment, and aerospace to solve electromagnetic radiation problems [4,5].

Compared to carbonyl iron powder particles of other shapes, layered carbonyl iron powder (CIP) possesses a higher specific surface area, aspect ratio, shape anisotropy, magnetic permeability, and Curie temperature, and hence can overcome Snooke’s limit and provide significantly better absorption performance. Additionally, microwave, having the advantages of high magnetic permeability, small matching thickness, good temperature stability (Curie temperature up to 770 °C), and high saturation magnetization, has broad application prospects and potential for development in the field of stealth technology, thereby receiving increasing attention [6,7]. CIP composite-absorbing materials were prepared using epoxy–silicone resin as the matrix, and the influence of the absorbent content on the absorption properties of the composite materials was studied. The lowest reflection loss of the composite materials was observed at −42.5 dB, and the composite materials showed good mechanical properties [8]. Other studies have uniformly mixed CIP with different matrix materials, such as epoxy resin [9], rubber [10], chloroprene rubber [11], and polyurethane [12], to prepare composite materials and studied the relationship between the mass fraction of CIP and the absorbing properties of the composite materials. However, due to its high density, CIP settles rapidly during the mixing process with resin, significantly affecting the process stability in practical applications.

Magnetic particles often need to be modified to achieve excellent application performance. For example, magnetic particles are relatively dark, making them difficult to use as anti-counterfeiting materials in light-colored media. By coating their surface with titanium dioxide and silver, they become light-colored magnetic particles, thereby expanding their applications [13]. Surface modification of materials can significantly improve their application performance as well; for example, modifying the surface of steel plates is seen to enhance their adhesive strength [14].

Fillers are essential additives in resin coatings to meet special requirements, but the settling stability of most density fillers seriously affects the stability of their construction process. For example, the density of magnetic materials themselves is very high, and the problem of dispersion and settling in resins is serious, which makes it difficult for magnetic materials to be evenly dispersed in coatings. So, in order to solve the dispersion problem of high-density materials in liquid resins, this paper proposes a solution.

Hollow silica is a novel inorganic material [15]. The particle has a size between 10 and 30 mm and density of about 0.5 g cm^−3^. It consists of a hollow structure surrounded by a wall with a thickness of about 20 nm. Hollow silica microspheres have been widely used as carriers in fields such as medicine, catalysis, energy, and the environment owing to their large specific surface area, high chemical stability, and low density. The silica surface contains a large number of silicon hydroxyl groups, which can be easily chemically bonded to modify its surface, further expanding its application range.

N-[3-(Trimethoxysilyl)propyl]ethylenediamine (KH792) was grafted onto the surface of the magnetic particles through a chemical bonding reaction, forming surface coated magnetic particle cations with amino groups. The pH value of 1 g of modified magnetic particles in 10 mL of aqueous solution is 9.28, indicating an anionic surface. The hollow silica surface was coated with carboxyl groups via vinyl silane bonding and acrylic polymerization, forming hollow silica anionic microspheres. The pH value of 1 g of SiO_2_ coated with polymer in 10 mL of aqueous solution is 4.65, indicating a cationic surface. The two materials were then mixed in specific proportions to obtain a low-density magnetic composite material. The density of composite magnetism can be adjusted according to the ratio between hollow silica and magnetic particles. This was found to greatly reduce the density of the magnetic particles, allowing them to be evenly dispersed in the resin solution, thereby providing strong support for ensuring product quality.

## 2. Experimental Section

### 2.1. Experimental Reagents and Instruments

The chemical reagents used to reduce the density of the magnetic materials are listed in Table 1, and the instruments and equipment involved in Table 2.

### 2.2. Surface Modification of Magnetic Particles

The surface of the CIP was oxidized in air. A large number of iron hydroxyl groups were found to be present on the surface, which, in turn, act as active sites and bond amino groups to CIP through the silanization reaction. A process diagram of the surface modification of the CIP is shown in Figure 1.

Briefly, 200 g of CIP was placed in a 500 mL three-necked flask, to which 250 mL of toluene was added. A reflux condenser tube with a water separator was attached to the flask at the mouth, heated, and further allowed to reflux under mechanical stirring for 2 h. Then, 100 mL of KH792 was added to the reaction mixture dropwise using a dropping funnel, during which 50 mL of the liquid was slowly released from the water separator. The reaction mixture was subsequently allowed to reflux for 4 h. After cooling, the solution was filtered and washed thrice with toluene and ethanol to obtain a CIP surface coated with amino groups.

### 2.3. Surface Modification of Hollow Silica

Figure 2 shows a flowchart of the surface modification process for the hollow silica. Ethylene groups are bonded to the surface of the hollow silica through a silanization reaction and further polymerized with acrylic carboxylic monomers to fix the carboxyl groups. The specific reaction process is illustrated in Figure 2.

Next, 40 g of hollow silica microspheres was placed in a 500 mL three-necked flask, to which 250 mL of toluene was added. A reflux condenser tube with a water separator was attached to the flask at the mouth, heated, and further allowed to reflux for 2 h under mechanical stirring. Following this, 80 mL of vinyltrimethoxysilane was added to the reaction mixture dropwise using a dropper funnel, during which 50 mL of the liquid was slowly drained from the water separator. The reaction mixture was further allowed to reflux for 4 h after the dropwise addition was complete. After cooling, the mixture was filtered and washed with toluene and ethanol at least thrice to obtain hollow silica with surface-bonded vinyl groups.

Then, 20 g of ethylene based hollow silica was placed in a 500 mL three-necked flask, to which 1.5 mL maleic acid and 5 mL acrylic acid were added. N_2_ gas was introduced under mechanical stirring for 30 min, following which 0.18 g of azobisisobutyronitrile was added. The reaction was allowed to continue for 4 h at 70 °C before being allowed to cool. After cooling, the samples were filtered and washed with toluene, acetone, and ethanol to obtain hollow silica microspheres coated with carboxyl groups.

### 2.4. Preparation of Low-Density Magnetic Particles

Mix a certain amount of amino modified magnetic particles with varying amounts of surface carboxyl modified hollow silica microspheres to obtain low-density magnetic materials of different densities. The amino groups on the surface of the magnetic particles and the carboxyl groups on the surface of the hollow silica particles form stable magnetic composite materials through chemical reactions and charge attraction. Furthermore, the density of magnetic composite materials can be adjusted by adding the hollow silica particles.

## 3. Results and Discussion

### 3.1. Surface Modification of Hollow Silica

The surface morphologies of the hollow silica microspheres before and after modification were studied using scanning electron microscopy (SEM). Figure 3(A1,A2) and Figure 3(B1,B2) show the SEM morphology of the hollow silica microspheres before and after modification, respectively. On comparing the figures, it can be seen that the surface was smooth before modification and shows obvious polymer protrusions after the modification, indicating that the polyacrylic acid substance coated the hollow silica surface.

Figure 4 shows the SEM morphology of silica coating modified with different amounts of polymer monomers while maintaining the same weight of silica. As the amount of the acrylic monomer added to the polymerization coating reaction system increases, the polymer coating on the surface of the silica becomes thicker. When the ratio of polymerized monomers to silica was 30%, the coating thickness was relatively high (Figure 4B), while still being well dispersed as individual particles. At higher concentrations of the polymerized monomers, adhesion occurred between the silica spheres (Figure 4C), leading to particle aggregation and the inability to form a good dispersion system, while at lower concentrations, the polymer coating layer was relatively thin (Figure 4A). A very thin layer of the polymer coating can lead to a decrease in the stability of the magnetic composite material. Therefore, to obtain a stable magnetic composite material, the polymer coating layer must attain a certain optimum thickness.

Figure 5 shows the infrared spectra (IR) of the hollow silica microspheres before and after coating, where the red and black curves represent the modified and unmodified samples, respectively. A significant difference could be observed in IR between the two groups. The IR of the coated hollow silica microspheres shows significant CH stretching vibration absorption at 2900.88 cm^−1^, with a considerable increase in hydroxyl vibration absorption at 3436.68 cm^−1^. At the same time, carbonyl vibration absorption appears at 1723.51 cm^−1^, confirming the coating of the carboxylic acid polymers on the silica surface.

In this approach, 1 g each of the polymer-coated and uncoated SiO_2_ samples were taken, 10 mL of deionized water was added, and the samples were shaken thoroughly. The pH was measured using a pH meter. The pH of the uncoated SiO_2_ aqueous solution was 6.65 ± 0.02, while the pH of the SiO_2_ aqueous solution coated with polymer was 4.65 ± 0.02. This indicated that carboxylic acid polymers had been coated onto the surface of the SiO_2_.

### 3.2. Characterization of Modified CIP

The CIP of surface-bonded silane coupling agents (KH792) has limited characterization methods due to the presence of only one layer on the surface, with a thickness of less than 1 nanometer. IR cannot be detected. The spatial resolution of SEM-EDS is too low to detect. X-ray photoelectron spectroscopy (XPS) can detect the content of elements within a depth of 5 nanometers, so it can barely be used in analytical detection. Table 3 shows the XPS of the unmodified CIP and modified CIP. The presence of Si and N in the unmodified CIP was also measured, which was attributed to experimental errors in the instrument. C was caused by interference from carbon dioxide, while O had an oxide layer on the surface of the CIP. XPS measurements showed high contents of Si, C, N, and O in the modified CIP, indicating that KH792 had bonded to the surface of the CIP.

Next, 1 g of modified and unmodified CIP samples were taken, 10 mL of deionized water was added, and the samples were shaken thoroughly; then, the pH was measured with a pH meter. The pH of the unmodified CIP aqueous solution was 6.65 ± 0.02, while the pH of the modified CIP aqueous solution was 9.28 ± 0.02. This indicated that KH792 had bonded to the CIP surface, forming an anionic surface.

### 3.3. Morphological and Structural Characterization of Magnetic Composite Materials

Figure 6 shows the SEM image of magnetic composite materials. It can be observed that sheet-like carbonyl iron particles with a shape of 5 μm × 400 nm are coated on the surface of silica spherical particles with a diameter of about 20 μm. This confirms that the amino groups on the surface of the magnetic particles reacted with the carboxyl groups on the surface of the spherical SiO_2_ to form stable chemical bonds, which firmly adsorbed the magnetic particles onto the hollow silica surface to form stable low-density magnetic composite materials.

Figure 7 shows the elemental surface distribution in the SEM energy spectrum (SEM-EDS) of the magnetic composite material. It can be seen that the sheet-like magnetic particles are uniformly adsorbed onto the spherical SiO_2_ surface. The distribution of iron elements is in the shape of sheet-like carbonyl iron, whereas Si and O are in the shapes of SiO_2_.

Figure 8 shows the X-ray diffraction (XRD) patterns of the CIP before (black) and after (red) modification. The diffraction curves show a stable body-centered cubic crystal structure and space group Fm-3m (international code 229), indicated by the anatase standard card (JCPDS card number PDF#06-0696). Owing to the amorphous nature of the hollow silica microspheres and the absence of X-ray diffraction absorption, there was almost no change in the XRD patterns before and after the CIP modification of the composite.

### 3.4. Characterization of Suspension Properties of Magnetic Composite Materials

The dispersibility of the material in the resin liquid is significantly affected by the density of the material. Therefore, reducing the density of a material is essential for improving its process stability. A conventional density testing method is the volume exclusion method; however, it is difficult to measure the density of porous particle samples accurately. Therefore, in this study, the stacking volume method was used to determine the relative density of the material. In this test, a certain amount of material was taken in a measuring cylinder and shaken repeatedly until no further change was observed in the volume scale inside the cylinder. From the scale in the cylinder, the accumulated volume of the material was then obtained. The lower the density of the material, the larger the stacking volume. Different weights of carbonyl iron powder were mixed with hollow silica particles to form composite materials. For conducting this experiment, 100 g of the composite material was taken in a graduated cylinder and shaken repeatedly. From the corresponding reading observed on the graduated cylinder, the stacking volume of the composite materials (mL/100 g) was obtained. Figure 9 shows the stacking volume of a magnetic composite material composed of CIP and hollow silica nanoparticles. The higher the proportion of CIP in the composite material, the lower the stacking volume and thereby the higher the density of the magnetic composite material. The stacking volume of 100 g of pure CIP was 48 mL, while that of 100 g of the magnetic composite material reached 218 mL, where CIP accounted for 66.7%.

CIP has a relatively high density and settles in the resin solution over time. The settling speed of the CIP was measured by uniformly mixing it into the resin solution and measuring the liquid–solid content of the upper resin after a certain interval of time. Accordingly, the CIP resin liquid was uniformly mixed, and the solid content of the upper layer of the resin liquid after a certain interval of time was used to evaluate the sedimentation of the magnetic material. Figure 10 shows the ratio of the liquid–solid content of the upper layer of the magnetic material resin solution after a 30 min settling period to its initial solid content. Initially, the solid content of the resin solution was approximately 52%, and the proportion of magnetic material added to the resin solution was 50%. Therefore, the initial solid content of the magnetic resin solution was 76%. Before modification with CIP, the supernatant of the magnetic material resin solution after being left for 30 min had almost no CIP, and the solid content was approximately 52%. Therefore, the ratio of the solid content of the supernatant of the magnetic material resin solution after being left for 30 min to its original solid content indicated a solid content retention rate of approximately 68%. As the proportion of silica in the composite magnetic materials increased, the CIP content was found to decrease, while the retention rate of the solid content in the upper layer of the magnetic material resin solution increased after being left for 30 min. At 70 wt% of CIP in the magnetic composite material, the retention rate of the upper layer gel solid content after 30 min was found to be close to 100% (Figure 10).

## 4. Conclusions

Materials with hydroxyl groups on their surfaces can be modified through silanization reactions to obtain functional materials with interfacial functional groups. Functional groups can also be grafted onto the surfaces of materials through polymerization reactions. Acrylic monomers are coated on the surface of hollow SiO_2_ through polymers to form a cationic surface with a pH value of 4.65. By bonding KH792 to CIP, an anionic surface with a pH of 9.28 was formed. Afterwards, low-density polymer coated hollow SiO_2_ anions and high-density modified CIP cations were combined to form magnetic composite materials.

The settlement problem of high-density materials is a common issue in solution–material construction, and reducing and slowing down settlement is an important way to improve the stability of high-density material engineering. This communication indicates that high-density materials can be reduced in density by compounding with hollow SiO_2_, achieving the goal of reducing settling or no settling at all in liquid resins. By modifying CIP materials, the goal of reducing the density of magnetic materials was achieved, and the process stability of magnetic material applications was improved.

## Figures and Tables

**Figure 1 materials-18-03469-f001:**
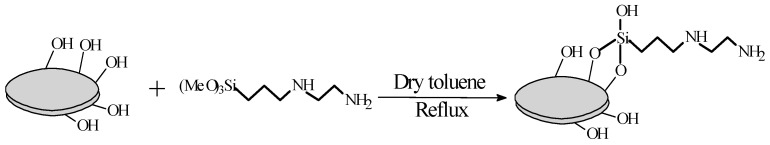
Block diagram of surface modification of magnetic particles.

**Figure 2 materials-18-03469-f002:**
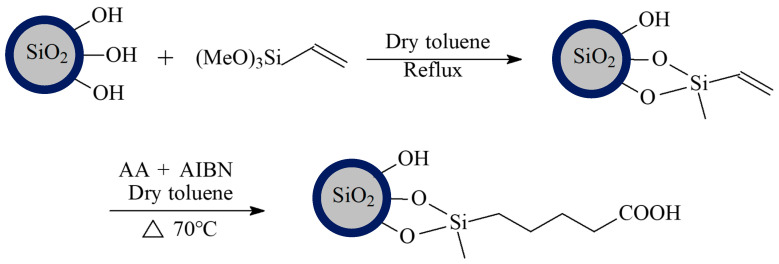
Block diagram of the surface modification of hollow silica.

**Figure 3 materials-18-03469-f003:**
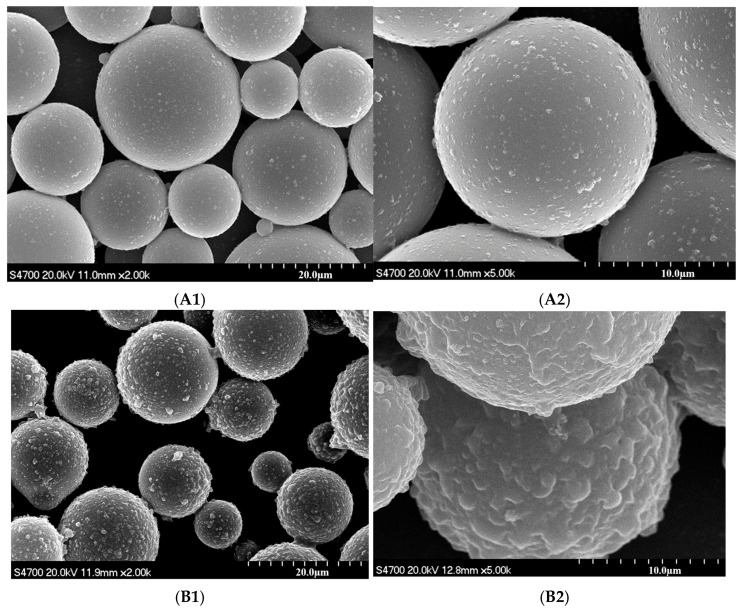
SEM images of hollow silica before (**A1**,**A2**) and after coating (**B1**,**B2**).

**Figure 4 materials-18-03469-f004:**
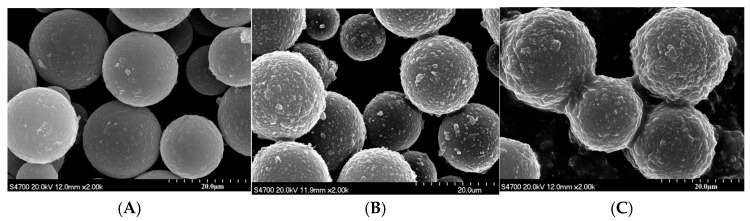
SEM morphology of silica coating modified with different amounts of polymer monomers, where (**A**), (**B**), and (**C**) represent the monomer addition ratios of 10%, 30%, and 50%, respectively.

**Figure 5 materials-18-03469-f005:**
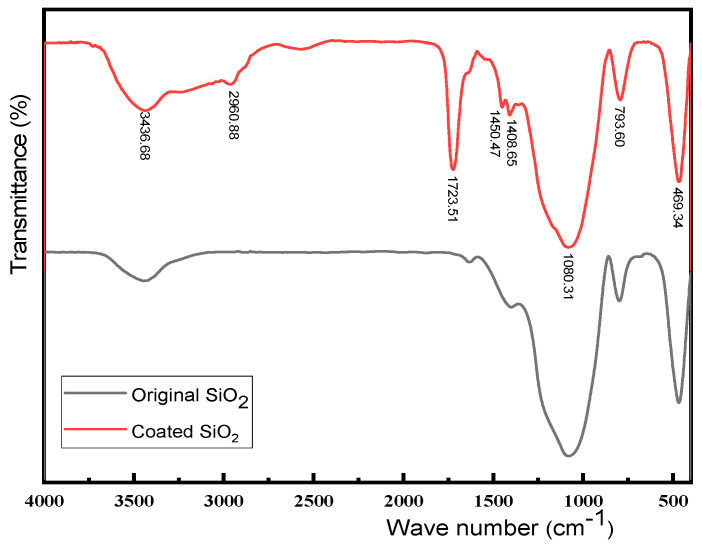
IR spectra of hollow silica microspheres before (black) and after (red) coating.

**Figure 6 materials-18-03469-f006:**
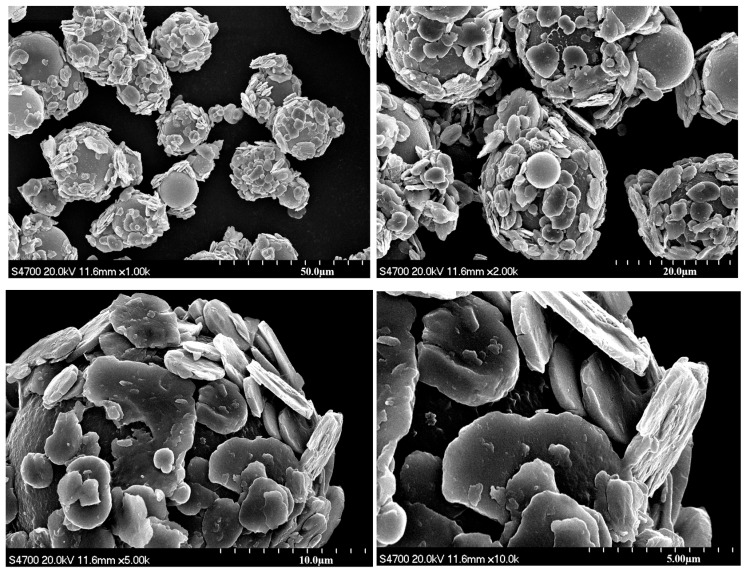
SEM images of magnetic composite material.

**Figure 7 materials-18-03469-f007:**
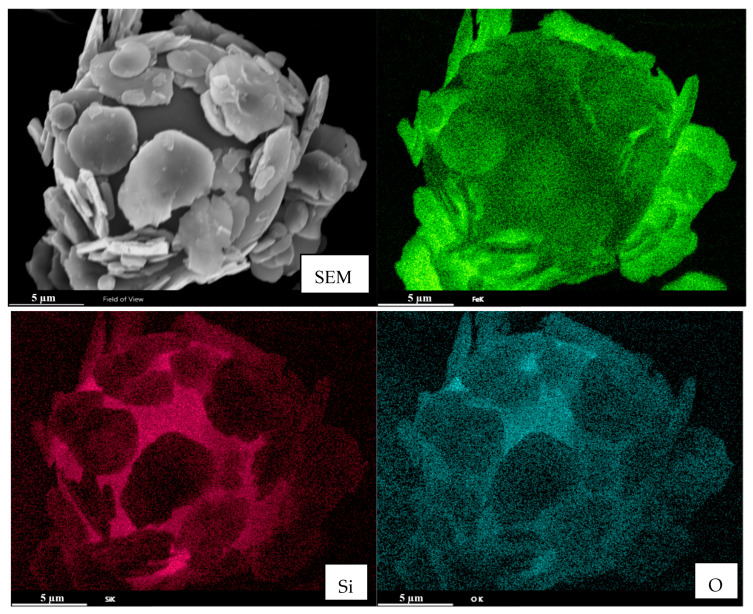
SEM-EDS of magnetic composite materials.

**Figure 8 materials-18-03469-f008:**
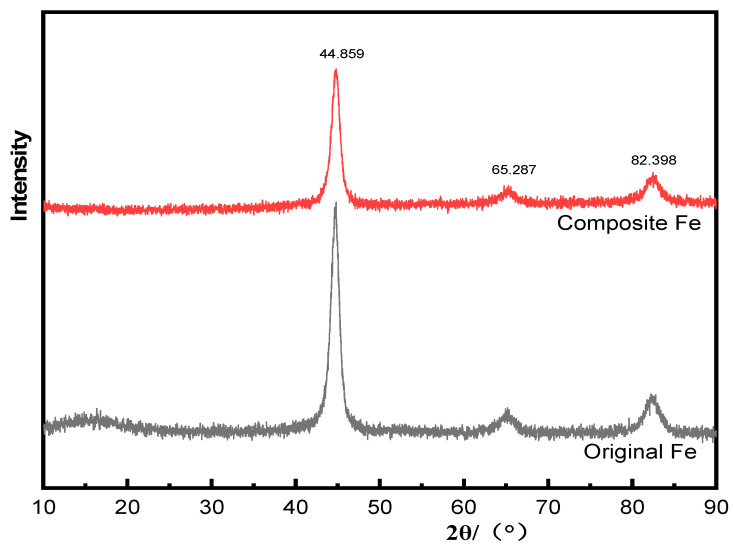
XRD patterns of CIP before (black) and after (red) modification.

**Figure 9 materials-18-03469-f009:**
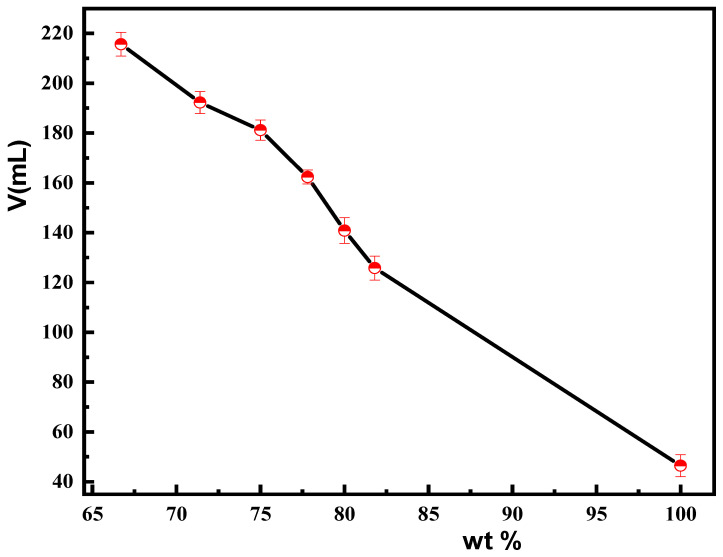
Stacked volume diagram of 100 g of magnetic composite materials with different proportions of CIP.

**Figure 10 materials-18-03469-f010:**
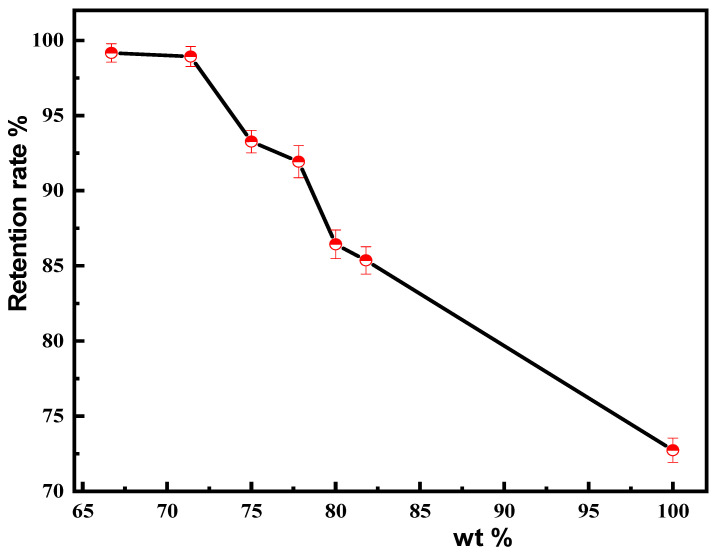
Retention rate of the upper solid content in magnetic resin solutions of magnetic composite materials with different CIP contents after 30 min.

**Table 1 materials-18-03469-t001:** Main experimental reagents.

Drug Name	Chemical Formula	Purity	Manufacturer
Carbonyl iron	Fe	99.95%	Henan Fluorine-based New Materials Industry Research Institute, Zhengzhou, China
Hollow silica	SiO_2_	99%	3M Company, Maplewood, MN, USA
N-[3-(Trimethoxysilyl)propyl]ethylenediamine	NH_2_CH_2_CH_2_NHCH_2_CH_2_CH_2_Si(OCH_3_)_3_	Analytical pure	Shanghai Aladdin Biochemical Technology Co., Ltd., Shanghai, China
Vinyltrimethoxysilane	CH_2_=CHSi(OCH_3_)_3_	Analytical pure	Shanghai McLean Biochemical Technology Co., Ltd., Shanghai, China
Toluene	C_7_H_8_	Analytical pure	China National Pharmaceutical Group Chemical Reagent Co., Ltd., Shanghai, China
Acetone	CH_3_COCH_3_	Analytical pure	Beijing Chemical Plant Co., Ltd., Beijing, China
Maleic acid	HO_2_CCH=CHCO_2_H	Analytical pure	Shanghai McLean Biochemical Technology Co., Ltd., Shanghai, China
Acrylic acid	CH_2_=CHCO_2_H	Analytical pure	Shanghai Aladdin Biochemical Technology Co., Ltd., Shanghai, China
Azobisisobutyronitrile	C_8_H_18_N_4_	Analytical pure	Shanghai Aladdin Biochemical Technology Co., Ltd.
Ethanol	CH_3_CH_2_OH	Analytical pure	Beijing Chemical Plant Co., Ltd., Beijing, China

**Table 2 materials-18-03469-t002:** Main experimental instruments and equipment.

Experimental Instrument Name	Instrument Model	Producer
Electronic balance	AL204	Mettler-Toledo International Trading (Shanghai) Co., Ltd., Shanghai, China
pH meter	FE28	Mettler-Toledo International Trading (Shanghai) Co., Ltd. China
CNC ultrasonic cleaner	KQ-400DB	KunShan Ultrasonic Instruments Co., Ltd., Kunshan, China
Electric blast constant temperature	101-0B	Shaoxing Yuanmore Machine and Electrical Equipment Co., Ltd., Shaoxing, China
Field emission scanning electron microscope	HITACHI S-4700	Hitachi, Tokyo, Japan
Fourier Transform Infrared Spectroscopy	Nicolet8700	Thermo Nicolet Co., Ltd., Madison, WI, USA
X-ray diffractometer	XRD-6000	Shimadzu, Kyoto, Japan
X-ray photoelectron spectrometer	ESCALAB 250	ThermoFisher Scientific, Waltham, MA, USA

**Table 3 materials-18-03469-t003:** XPS for unmodified and modified CIP.

Element Name	Unmodified CIP (wt%)	Modified CIP (wt%)
Si	0.25	4.71
C	5.67	20.38
N	0.17	3.21
O	12.89	25.46
Fe	81.02	46.24

## Data Availability

The original contributions presented in this study are included in the article. Further inquiries can be directed to the corresponding author.

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
