# Peer review of "Research on Preventing High-Density Materials from Settling in Liquid Resin"

_materials, 2025, doi:10.3390/ma18153469_

Round 1

Reviewer 1 Report

Comments and Suggestions for Authors

The authors presented a method for reducing the density of high-density magnetic particles. The results obtained may be very interesting, but the presentation of the material raises a number of questions and comments.

1) What is the purpose and novelty of the work? Why are such studies needed? These explanations should be inserted into the main text of the article.

2) Why was IR analysis performed only for SiO2 before and after modification and not for the final composite? Similar question: why are XRD data for CIP before and after modification presented and not for the resulting composite?

3) What conclusion can be drawn from the data of the X-ray image of the CIP?

4) Are there any magnetic measurements of the CIP? If this has been done in earlier works, please cite it.

5) What conclusion can be drawn from the data in Figures 9 and 10?

The Communication in the presented form cannot be accepted for publication. The article is "raw". It is clear from the text of the article that there are interesting results, but they were presented without proper discussion. Conclusions should be made as a conclusion on the work, on the results obtained, and not a repetition of the problem statement with a statement of the results obtained.

Author Response

The reviewer's response is attached.

Reviewer 2 Report

Comments and Suggestions for Authors

I have gone through the article “Research on Preventing High Density Materials from Settling in Liquid Resin” (ID: materials-3725695).

In this work, the authors report the synthesis of high density magnetic particles coated onto the surface of hollow silica using anion-cation composite technology. Also, the surface chemistry of the particles was modified using N-(β-aminoethyl)-γ-aminopropyltrimethoxysilane (KH792) in combination with the polymerization of acrylic acid to form a carboxyl-covered interfacial layer.

This article seems interesting for the synthesis and chemical functionalization of silica particles with the aim to solve the problem of magnetic particles sedimentation in the resin solution.

Some minor points need to be addressed:

  • Abstract: the full name of the chemical compound (KH792) must be specified the first time it is mentioned.
  • N-(β-aminoethyl)-γ-aminopropyltrimethoxysilane can be written as N-(β-aminoethyl)
  • Some keywords are in uppercase and others in lowercase, check the journal's style.
  • Table 1: “isobutyronitrile C8H18N4” (C8H18N4).
  • Figure 2: Did the authors use a chemical software to draw the molecules? (ChemDraw or something similar?).
  • Both axes in the figures can be corrected for an appropriate presentation of the results.
  • Check the references styles, g. [1.].

Some major points need to be addressed:

  • Regarding chemical aspects, the authors need to show all the chemical processes using FT-IR, X-ray Photoelectron Spectroscopy (XPS), or solid-state Nuclear Magnetic Resonance (NMR) in order to demonstrate what there are saying in the manuscript.
  • Moreover, the Inductively Coupled Plasma Mass Spectrometry (ICP-MS) or organic elemental analysis can be done in all the samples with the aim of enabling reproducibility in the synthesis technique.
  • SEM-EDAX studies was only done in one sample, what about the others?
  • The density of carboxylic acid residues is not specified. This information can be obtained from the potentiometric titration of the particles.
  • The conclusion section must be improved.

While the article is interesting, it lacks an appropriate chemical analysis/characterization.

Author Response

The reviewer's response is attached.

Round 2

Reviewer 1 Report

Comments and Suggestions for Authors

The authors have taken into account all the reviewer's comments. Thanks.

Reviewer 2 Report

Comments and Suggestions for Authors

All the previous points/questions have been addressed in the new version of the manuscript. This work can now be considered for publication.